# Beyond the Limits of Light: An Application of Super-Resolution Confocal Microscopy (sCLSM) to Investigate Eocene Amber Microfossils

**DOI:** 10.3390/life13040865

**Published:** 2023-03-23

**Authors:** Dmitry D. Vorontsov, Vasiliy B. Kolesnikov, Elena E. Voronezhskaya, Evgeny E. Perkovsky, Marielle M. Berto, Joseph Mowery, Ronald Ochoa, Pavel B. Klimov

**Affiliations:** 1Koltzov Institute of Developmental Biology, Russian Academy of Sciences, 119334 Moscow, Russia; 2Federal Public Budgetary Scientific Institution, All-Russian Research Institute of Plant Protection, 396030 Voronezh, Russia; 3Institute of Environmental and Agricultural Biology (X-BIO), University of Tyumen, 625003 Tyumen, Russia; 4Department of Entomology and Collection Management, I.I. Schmalhausen Institute of Zoology NAS of Ukraine, 01030 Kiev, Ukraine; 5Natural History Museum of Denmark, 2100 Copenhagen, Denmark; 6Tropical Research and Education Center, Homestead, FL 33031, USA; 7Agricultural Research Service, USDA, Beltsville, MD 20705, USA; 8Systematic Entomology Laboratory, Agricultural Research Service, USDA, Beltsville, MD 20705, USA; 9Department of Biological Sciences, Purdue University, West Lafayette, IN 47907, USA

**Keywords:** fossil, Acari, Acaridae, super-resolution confocal microscopy, fluorescence, Eocene

## Abstract

Amber is known as one of the best sources of fossil organisms preserved with exceptional fidelity. Historically, different methods of imaging have been applied to amber, including optical microscopy and microtomography. These methods are sufficient to resolve millimeter-scaled fossils. However, microfossils, such as microarthropods, require another resolution. Here, we describe a non-destructive method of super resolution confocal microscopy (sCLSM) to study amber-preserved microfossils, using a novel astigmatid mite species (genus *Histiogaster*, Acaridae) from Eocene Rovno amber as a model. We show that the resolution obtained with sCLSM is comparable to that of scanning electron microscopy (SEM) routinely used to study modern mites. We compare sCLSM imaging to other methods that are used to study amber inclusions and emphasize its advantages in examination of unique fossil specimens. Furthermore, we show that the deterioration of amber, which manifests in its darkening, positively correlates with its increased fluorescence. Our results demonstrate a great potential of the sCLSM method for imaging of the tiniest organisms preserved in amber.

## 1. Introduction

Amber-embedded fossils provide a unique snapshot of the evolution on our planet because they are preserved with great fidelity and are relatively common. The oldest amber-preserved arthropod specimens are known from the Upper Triassic, ca. 230 Mya [1]. There are more than 150 known amber deposits around the world [2,3], while only a few of the most studied, such as Cretaceous Kachin amber (99 Mya), Eocene Baltic and Rovno amber (37–34 Mya) and Miocene Dominican amber (20–15 Mya) provide the bulk of the reported biodiversity [4].

Morphological resolution of amber fossils is often comparable with that of extant specimens, as was demonstrated by transmission electron microscopy (TEM) in thin sections of amber, where even the cellular fine structures could be resolved [5]. Studies of larger fossils, such as arthropods of several millimeters in size, usually do not require to be imaged at such a high resolution [6,7], although in some cases, a micrometer-scale resolution is needed to study important diagnostic characters [8]. In contrast, fossil microarthropods, and mites (Acari) in particular, always require high-resolution imaging methods to reveal their morphology in the finest detail.

Traditionally, modern microarthropods are studied with the use of light microscopy and scanning electron microscopy (SEM), both providing sub-micron resolution. For amber-preserved specimens, light microscopy has been used since the 18th century [9]; however, its success largely depends on the opacity of amber specimens. The resolution of light microscopy can be significantly increased by fine amber preparation or polishing, so the biological specimen is within the distance of higher magnification objectives [10,11]. The resolution of transmitted light microscopy can be further improved in combination with immersion methods [12] that hide imperfections of polishing and decrease the scattering of light. However, in terms of resolution, this method is still inferior to SEM, which is routinely used to reveal fine morphology in modern arthropods. Unfortunately, SEM is hardly applicable to amber fossils, as it requires the free surface of the specimen to be exposed for scanning. Several attempts have been made to extract fossils for SEM imaging after the amber pieces have been intentionally or occasionally fractured [13,14,15,16,17]. Although this approach often provides unique information, with amber fossils, both SEM and TEM are destructive methods. The use of these methods is not encouraged by the curators of museum collections and these methods are rarely applied to unique museum specimens.

The method of X-ray microscopic computed tomography (μ-CT, nano-CT) and especially, the propagation phase contrast synchrotron X-ray microtomography (PPC SRμCT) has proved to be indispensable for opaque amber pieces [18,19,20,21,22]. Microtomography allows revealing the internal morphology of a fossil which may be otherwise hidden [23]. It has been applied to modern [24] and fossil mites [25]. However, although the spatial resolution currently available for nano-CT and SRμCT, ca. 100 nm [26], is enough to study fossil microarthropods, the practical limit of resolution achieved in the studies of amber fossils often does not allow to reveal taxonomically important characters [27,28].

Confocal laser scanning microscopy (CLSM) is another non-destructive method suitable for studies of amber fossils. Its usage is based on the effect of autofluorescence demonstrated by the fossilized exoskeletons of arthropods [29,30]. Unlike traditional light microscopy, CLSM rejects most out-of-focus light and allows to obtain three-dimensional images of the specimens. It was successfully applied to different amber fossils, including the studies of fossil spider web [31] and filamentous mycelium [16,32] performed with high resolution in thin sections of amber (see [33] for review of the CLSM application to amber fossils).

We recently applied super-resolution confocal laser scanning microscopy (sCLSM) in the study of fossil mites from Eocene and Cretaceous amber [34]. Super-resolution microscopy bypasses the diffraction limit, a physical barrier that restricts the optical resolution to roughly 250 nm. The method that we used, Zeiss Airyscan [35], is based on a computation of data obtained from the array of 32 detectors, each functioning as a very small pinhole to increase the resolution of imaging, while the whole array delivers better signal-to-noise ratio than traditional confocal systems. In that study, we confirmed that the state of preservation in amber fossils may allow excellent imaging resolution exceeding that of light microscopy. However, sCLSM in fossils has not yet been evaluated in comparison with SEM imaging of modern animals. In addition, it is still not clear what the reason is for decreased fluorescence in some fossils that often occurs along with intense fluorescence of surrounding amber [30] which prevents successful CLSM imaging. We hypothesized that autofluorescence of amber may correlate with its deterioration, which manifests as darkening or yellowing of amber and reflects the process of its depolymerisation [36].

In this study, we set an ambitious goal to achieve the resolution in amber fossil mites comparable to that of SEM, while taking full advantage of the non-destructive approach provided by sCLSM. We study mites of the genus *Histiogaster* that lived in tree sap flow and were uniquely preserved in Eocene amber mined in Klesov, Rovno Oblast, Ukraine (the biota of the deposit was reviewed in [37]). The mite species belongs to the genus *Histiogaster* (we present the first fossil record of the genus) and has only minor differences in comparison with extant species. We can therefore confidently compare the fossil specimens with modern mites of the same genus. In *Histiogaster*, species delimitation is based on minute characters, such as leg setae and the shape of the tip of the Grandjean’s organ, which is only a few micrometers in size. We used sCLSM to resolve these minute details and compare our results with images taken by low-temperature SEM (LT-SEM) of modern *Histiogaster* mites as a reference. 

## 2. Materials and Methods

### 2.1. Fossil Material

Two pieces of Eocene Rovno amber containing mites were taken for study from the amber collection of the Schmalhausen Institute of Zoology, NAS of Ukraine, Kiev (SIZK), collection numbers SIZK K-15219 and SIZK K-15220. Both pieces were cut from the original piece number 3-694 (weight 10.3 g, size 40 × 34 × 15 mm) from the unbiased collection obtained at the “Ukramber” factory and mined at Pugach quarry (Klesov, Ukraine). The original pieces of amber were dark yellow (Figure 1a), with many areas almost opaque due to numerous microinclusions (Figure 1c), presumed to be a fossilized emulsion of a phloem sap. At the microscopic level, the amber appeared transparent. One of the amber pieces contained filamentous mycelium (Figure 1b,d), the detailed study of which is pending, that also suggested the presence of rich organic substrate such as a phloem sap. A total of 11 specimens (eight females and three males) of mites from the genus *Histiogaster* were found in different states of preservation. The description of the new species will be published separately. 

Initial reflected light images of fossil mites rendered the surface of mite inclusions distorted (Figure 1e,f), and even the major setae were hardly traceable against the contrast fractures. However, a closer look using CLSM after fine polishing of amber pieces closer to the mites revealed that many taxonomically important parts of fossils were intact (Figure 1g,h).

To study the correlation between the deterioration (yellowing) of amber and the level of its autofluorescence, we prepared several blocks of amber, cutting them perpendicular to the surface of the original pieces, which had noticeably darkened and deteriorated surfaces. Two blocks were cut from the same Rovno amber pieces where the mites were discovered (SIZK K-15219, K-15220); one of the blocks included two mite specimens. To test whether a similar correlation was present in different amber pieces, we cut several blocks from a piece of Rovno amber (collection number SIZK VT-305) mined at a different location, near the village of Velyki Telkovichi (Varash Distr., Rovno Oblast, Ukraine), and from the Eocene Baltic amber (collection of the Paleontological Institute, Moscow, specimens PIN 964-1139, 964-1140 and 964-1149).

### 2.2. Preparation of Amber

In order to achieve the maximum resolution of the CLSM and the high-resolution optics, we minimized the optical imperfections of the amber specimens by (i) fine-polishing amber pieces close to a fossil and (ii) using saturated fructose solution as the immersion medium between the amber piece and a coverslip. The pieces of amber were trimmed and polished as described by Sidorchuk and Vorontsov [11], which allowed to precisely control the thickness of amber between the fossil and the polished surface. If the use of high-resolution immersion optics was involved, amber was polished such as to leave not more than 100 μm (normally, ca. 50 μm) from the surface to the inclusion. Processed amber pieces were stored in water. For microscopic observation, pieces of amber were mounted in saturated fructose on one side of the coverslip, to be approached by the microscope objective from the other side of the coverslip (see [34] for details). For inverted microscopy, we used a custom ‘beetle house’ accessory [11], which is a 2–5 mm thick plastic microscope slide with a hole in it, sealed with a coverslip on one side. Alternatively, a cell imaging dish with a glass bottom can be used. It is important to dry the surface of amber before immersing it in fructose solution; otherwise, the formation of crystals may follow. In some cases, it is enough to apply a drop of saturated fructose onto the glass and then put the specimen over it with the side to be imaged down, gently pressing the amber piece with a toothpick to remove any bubbles between the glass and amber. In case of excessive quantity of fructose, the specimen may slip sideways during the imaging, which can be prevented by fixing it in place between two pieces of 1 mm-thick VHB sticky tape (3M) before applying the fructose solution.

In many cases, polishing of only two surfaces of an amber piece, for example, dorsal and ventral aspects of a fossil, was not enough. In the fossil *Histiogaster* mites studied here, their legs and gnathosoma, which have many diagnostic characters, were situated almost perpendicular to the initial fine-polished planes. This was in contrast to the traditional slide-mounting technique for modern mites, which usually flattens the specimens dorsoventrally so leg detail can be observed. To image these body parts in fossil mites, the amber specimens were polished in the shape of a parallelepiped, thus providing the lateral and frontal views in addition to the standard dorsal and ventral views.

### 2.3. Imaging of Fossils

Imaging in transmitted and/or reflected light was done using a Nikon E-800 and Zeiss AxioImager A2 compound microscope equipped with an Olympus OM-D E-M10-II digital camera. The following microscope objectives were used: Nikon E Plan 4×/0.10 (overview of amber pieces); Plan Fluor 10×/0.30, Plan Apo VC 20×/0.75, Fluor 40×/0.80 w, Plan Fluor 100×/1.30 oil (brightfield and differential interference contrast, DIC); Zeiss EC Plan Neofluar 100×/1.30 oil (DIC); Plan Achromat 10×/0.45 (regular CLSM), Plan Apochromat 40×/1.4 oil (regular CLSM); Plan Apochromat 63×/1.40 oil (regular CLSM and Airyscan).

Stacks of images comprising multiple focal planes were treated for color, digital noise and sharpness with Adobe Lightroom. Focus stacking, both automatic and manual (with retouching), was done in Helicon Focus 7.6.2. Confocal scanning microscopy was done using a Zeiss LSM-880 CLSM equipped with an Airyscan super-resolution detector. 

For imaging the amber fossils with CLSM, the choice of the appropriate wavelength of excitation is important. The exoskeletons of modern arthropods demonstrate autofluorescence when excited in blue and especially in the UV range. Endogenous fluorophores known for insect cuticle include resilin [38], chitin and chitinous compounds [39], and other substances [40]. However, a short wavelength elicits high background fluorescence of an amber matrix which masks the signal from a fossil. We used a 488 nm Argon laser for excitation; the detection range was set from ca. 500 to 700 nm in regular CLSM mode, while in sCLSM mode (Airyscan detector), a set of emission filters, BP 495–550 and LP 570, was used. These settings allowed the detection of green to yellow and red fluorescence. 

The parameter that mostly affected the quality of images was the dwell time (or pixel time, the longer the better). We found that autofluorescence in amber fossils is resistant to bleaching, so both the long dwell time and the high laser intensity may be used. We normally used dwell time around 2 μs, which gave ca. 10 s frame time for 1000 × 1000 px, 2 frames averaging, and ca. 30 min scan time for a 150–180-slice stack. For sCLSM scans, the z step was usually set to 0.2 μm. To view and manipulate the confocal image stacks, the original Zeiss Zen software, as well as a FIJI software package [35] and Helicon Focus software, were used.

To estimate the practically attainable resolution in sCLSM scans, maximum intensity projections were produced from the z-stacks acquired with pixel oversampling (normally, we used 2–3 times oversampling) and then downsampled with the ratios of 2, 3 or 4. The resulting images were visually compared side-by-side, and the resolution was estimated from the image where the loss of detail was not yet evident.

### 2.4. Modern Mites

Specimens of *Histiogaster arborsignis* Woodring, 1963 (Acaridae) were obtained from a stock culture maintained at the Department of Entomology and Nematology (University of Florida, Gainesville, FL, USA). The culture was established from three phoretic deutonymphs collected on ambrosia beetles. Mites were maintained in sealed autoclavable plastic bags (46 × 20 cm, height × diameter), on sterile moist rolled barley flakes at 25 ± 1 °C, 70 ± 10% R.H. The food source was changed every two weeks or whenever there was contamination with fungi or bacteria.

### 2.5. Imaging of Modern Mites

Specimens in 70% ethanol and/or live material were used for Low Temperature SEM (LTSEM) studies, utilizing the technique outlined by Bolton et al. [41]. Briefly, specimens were secured to 15 × 30 mm copper plates using ultra smooth, round (12 mm diameter) carbon adhesive tabs (Electron Microscopy Sciences, Inc., Hatfield, PA, USA). The specimens were frozen conductively in a Styrofoam box by placing the plates on the surface of a pre-cooled (−196 °C) brass bar whose lower half was submerged in liquid nitrogen. After 20–30 s, the holders containing the frozen samples were transferred to the Quorum PP2000 cryo-prep chamber (Quorum Technologies, East Sussex, UK) attached to an S-4700 field emission scanning electron microscope (Hitachi High Technologies America, Inc., Dallas, TX, USA). The specimens were etched inside the cryotransfer system to remove any surface contamination (condensed water vapor) by raising the temperature of the stage to −90 °C for 10–15 min. Following etching, the temperature inside the chamber was lowered below −130 °C, and the specimens were coated with a 10 nm layer of platinum using a magnetron sputter head equipped with a platinum target. The specimens were transferred to a pre-cooled (−130 °C) cryo-stage in the SEM for observation. An accelerating voltage of 5 kV was used to view the specimens. Images were captured using a 4 pi Analysis System (Durham, NC, USA).

## 3. Results

Low-resolution CLSM scanning demonstrated variable levels of autofluorescence in different specimens (*n* = 11): in some cases, a mite was very bright and contrasted against the dark background of amber (*n* = 7, example in Figure 1g); in others, the amber was highly fluorescent, almost masking the signal from the fossil (*n* = 2, example in Figure 1h). To understand such a diversity of the amber autofluorescence, we scanned blocks of amber that were cut perpendicular to the surface of the original pieces, comparing the fluorescence at different distances from the original surface (see below).

### 3.1. Autofluorescence Gradient in Amber Is Correlated with Amber Deterioration (“Yellowing”)

In the blocks of amber cut perpendicular to the deteriorated surface of the original piece, there was a clear gradient of yellowing: the color ranged from neutral in deeper layers of amber to rich yellow close to the surface of the original piece (Figure 2a,c,e,g,i). Scanning of the same blocks using a confocal microscope (488 nm excitation wavelength) revealed a correlated gradient of autofluorescence in amber: more yellow areas demonstrated higher fluorescence (Figure 2b,d,f,h,j). Besides the deteriorated surface of the piece, the internal darker borders between layers of amber possessed high levels of fluorescence (Figure 2h,j, arrows).

### 3.2. Comparing sCLSM with Transmitted and Reflected Light Microscopy

Figure 3 shows the sequential approach to one of the specimens: preliminary low-resolution imaging in unfavorable aspect after initial trimming of a mite from a larger piece (Figure 3a), trimming and polishing the six sides of the piece, including the frontal view to assess the gnathosoma (Figure 3b), imaging with dry 20× objective (Figure 3c,d), then imaging in saturated fructose immersion with regular CLSM (Figure 3e), then Airyscan sCLSM (Figure 3f–i). In several cases, confocal microscopy allowed to reconstruct the surface of a mite’s exoskeleton (Figure 3g,h). The time cost of imaging with sCLSM was not significantly higher than that of regular CLSM, provided that the pixel dimensions were similar. For example, scanning of the anterior part of a mite in regular CLSM mode (Appendix A) and the mouthparts of the same specimen at higher magnification in Airyscan mode (Appendix A) took ca. 30 min in both cases.

### 3.3. Estimating the Resolution of sCLSM

In our tests, the practically attainable lateral resolution of sCLSM in fossil mites was estimated to be 0.12–0.15 μm for a 488 nm laser, which corresponds to the theoretical limit for Airyscan CLSM [42]. It predictably surpassed the resolution of transmitted light microscopy (0.25 μm). Figure 4 shows Grandjean’s organs of fossil and modern *Histiogaster* mites. With transmitted light (100× oil immersion objective), the tip of Grandjean’s organ in the fossil mite can not be seen clearly (Figure 4a). The Airyscan CLSM method applied to the same fossil unequivocally resolved the bifurcated tip (Figure 4b). In the modern mite, the tip of Grandjean’s organ, although less bifurcated, was clearly resolved by LTSEM (Figure 4c,d).

Figure 5 compares legs and gnathosomal structures in fossil *Histiogaster* mites (sCLSM images, Figure 5a,d) and those of modern mites, *H. arborsignis* (imaged using LTSEM, Figure 5b,c,e), showing a similar level of detail in the fossil and modern mites.

## 4. Discussion

### 4.1. Ancient Habitat

Most living species of *Histiogaster* are known from subcortical habitats and have clear preference for fermented materials, such as tree sap, wine barrels, and commercially produced alcoholic beverages, such as pulque [43,44,45]. In our sample, we found numerous microinclusions, which were tentatively attributed to fossilized phloem sap (Figure 1c), and mycelium (Figure 1d). Filamentous mycelia projecting inward in the amber pieces was observed in different Cretaceous ambers, where they were identified as sheathed bacteria [16] or fungi [32,46,47] that consumed the phloem sap embedded in the resin. These data collectively suggest that the fossil *Histiogaster* mites inhabited (and most probably, fed on) a phloem sap flow of an angiosperm tree.

### 4.2. Fluorescence of Fossils and Amber

Darkening, or “yellowing”, is a sign of amber deterioration [36,48,49]. Experimental accelerated thermal ageing on representative Baltic amber samples showed that yellowing correlated with the breakdown of the amber polymer chains (depolymerisation) caused by oxidative radical reactions [36]. Cutting of an amber piece often shows a color gradient from the inner to the outer layers, which are susceptible to oxidation by atmospheric oxygen. As we found, the best ratio of signal (fluorescence of a fossil) to background (fluorescence of amber) could be achieved in deeper, less deteriorated and more transparent layers of amber when using the wavelengths considered optimal for imaging of arthropod fossils (see the Methods section). Relatively high levels of fluorescence of the amber matrix, however, does not always prohibit the high-resolution imaging of a fossil [46], as CLSM does not excite significant fluorescence in out-of-focus layers. When the specimen is imaged at high magnifications, the in-focus layer is very narrow; thus, the background fluorescence of amber masks the signal of a fossil to a lesser extent.

Our results confirm that amber deterioration in museum collections could be a serious problem which, in some cases, may not be detected with the ordinary light microscope. However, CLSM may serve as a sensitive method for the early diagnosis of the deterioration process. CLSM allows to perform the optical sectioning of the specimen alternative to its physical cutting to demonstrate the gradient of yellowing. However, without a calibrated workflow, only a pairwise comparison of amber pieces stored in different conditions can be performed.

Confocal microscopy presents a great advantage for imaging of the tiny morphological characters, including setae [38,50,51,52]. The applicability of CLSM is based on the ability of autofluorescence of the arthropods’ cuticle. In modern arthropods, this effect is based on the presence of resilin or chitinous compounds, with their respective blue or green-to-red autofluorescence, as well as of other fluorescent components [40].

While the exoskeleton fluorescence may look quite similar in modern and fossil arthropods [34] (Figure 3), the nature of fluorescence in the fossil exoskeletons remains unclear. In contrast to transmitted light microscopy, CLSM is insensitive to the presence of opaque structures inside fossils (e.g., the remains of internal organs). This feature allows to selectively image the external morphology of the fossil using CLSM (compare Figure 3c,e). At the same time, in some cases when the internal organs of a fossil were also preserved, CLSM allows to study them, since most of the out-of-focus light related to exoskeleton can be eliminated.

Fluorescent image stacks obtained by CLSM have another advantage over those taken using transmitted light microscopy. When using the light microscope, one needs to rely on semi-automated focus stacking, involving time-consuming manual retouching procedures (showing low-contrast structures of a fossil against the higher-contrast ‘debris’). At the same time, even the fully automated maximum intensity projection performed over the CLSM image stack often presents informative images (Figure 3g,h). In other cases, when brightly fluorescent structures obscure the view, a series of maximum projections over the parts of the image stack or even the source images can be later combined into the final image by the traditional focus stacking software (example in Figure 3i). In addition, it is quite easy to represent and measure confocal image stacks in three dimensions (Appendix A).

### 4.3. Resolution of sCLSM in Amber Fossils Compared to Other Methods

In the recent studies of amber fossils performed at the European Synchrotron Radiation Facility (Grenoble, France), the resolution of SRμCT, measured as a voxel size, could reach 0.212 μm [20]. This method allows to resolve even the fossils which do not possess an air-filled imprint (and thus a clear border of density). However, judging from the published images, thin setae projecting into the thickness of amber lack enough contrast of density with amber and could not be effectively imaged by SRμCT [20] (Figure 5) or a laboratory-based μCT scanner [53] (Appendix A). At the same time, the remains of cuticular structures that are still present in many amber fossils are effectively imaged using sCLSM with a resolution exceeding that of transmitted light microscopy. Setae and other thin structures, in most cases, appear as the brightest structures in the image (Figure 3e,i). Thus, CLSM and μCT can perfectly complement each other in the study of amber fossils, depending on the opacity of the amber. Three-dimensionality provided by μCT and CLSM opens possibilities for measurements that would be hardly possible for amber fossils using light microscopy. Furthermore, at the moment, confocal microscopes have become quite common and are more accessible than synchrotron facilities.

Based on our study, the following steps are required to achieve the full advantages of sCLSM in amber fossils: (i) fine polishing of amber close to the fossil to enable the usage of oil immersion optics with short working distances; (ii) immersion of amber in an appropriate medium, such as saturated fructose solution, which hides imperfections of even well-polished amber pieces, decreases the scattering of light on these imperfections and reduces light refraction on the surfaces of amber and of glass.

Our sCLSM scans of the fossil *Histiogaster* mites (Figure 4 and Figure 5) revealed most morphological features that are visible in the SEM images: the shape and position of setae on the legs (Figure 5a) and on the palps (Figure 5b). Since the estimated sCLSM resolution in our study approaches the limits of the method, we believe that even higher resolution can be achieved for our fossils with advancement of imaging technology. Our results demonstrate a great potential of the sCLSM method for resolving morphology of amber fossils in three dimensions at a spatial resolution currently inaccessible by other methods.

## Figures and Tables

**Figure 1 life-13-00865-f001:**
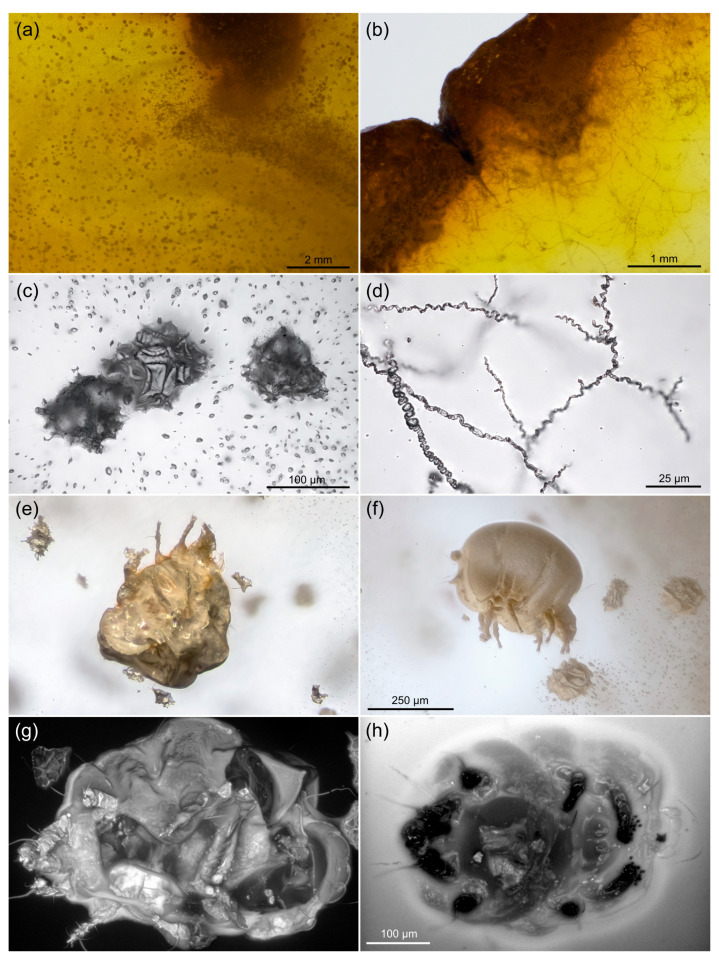
Different fossil inclusions found in the piece of Eocene Rovno amber. (**a**,**b**) Low-resolution micrographs of amber pieces showing the dark yellow color in thick slices, numerous microinclusions that fill most of the volume of amber and make it look misty. (**a**) The edge of the piece with filamentous mycelium (**b**); (**c**) Microinclusions, possibly fossilized tree sap; (**d**) Fragment of fossilized mycelium; (**e**,**f**) Preliminary images of fossil mites before final preparation; (**g**,**h**) Confocal scans of two fossil mites showing different levels of fluorescence in exoskeletons and in amber matrix surrounding them. (**e**,**f**) To same scale; (**g**,**h**) To same scale. (**a**–**d**) Transmitted light microscopy; (**e**,**f**) Combined transmitted/reflected light microscopy; (**g**,**h**) Confocal microscopy, 488 nm excitation laser, detection range from ca. 500 to 700 nm. All images are results of focus stacking.

**Figure 2 life-13-00865-f002:**
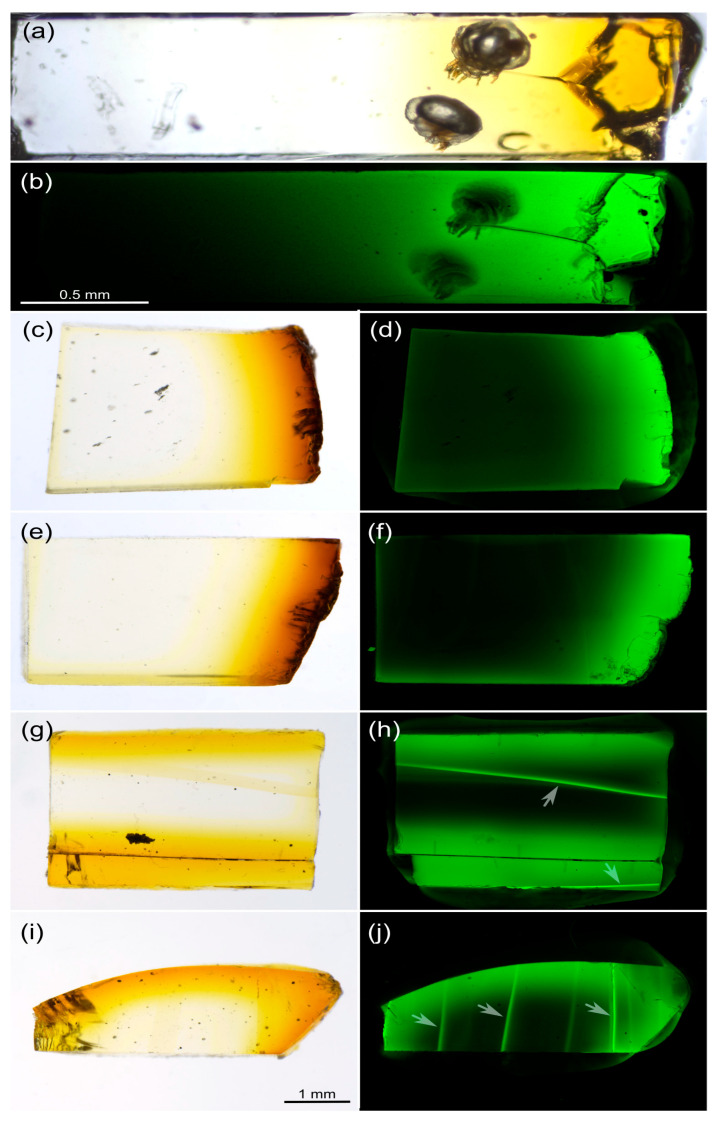
Correlation of amber deterioration (“yellowing”) with its autofluorescence. Blocks of amber cut perpendicular to the deteriorated surface of original pieces and imaged in transmitted light and in fluorescent mode (488 nm excitation laser). All pieces demonstrate a correlated gradient of yellowing (**a**,**c**,**e**,**g**,**i**) and fluorescence of amber (**b**,**d**,**f**,**h**,**j**). The piece shown in (**a**,**b**) contains two specimens of fossil *Histiogaster* mites. Note that the internal borders of the amber layers also demonstrate high levels of fluorescence (arrows).

**Figure 3 life-13-00865-f003:**
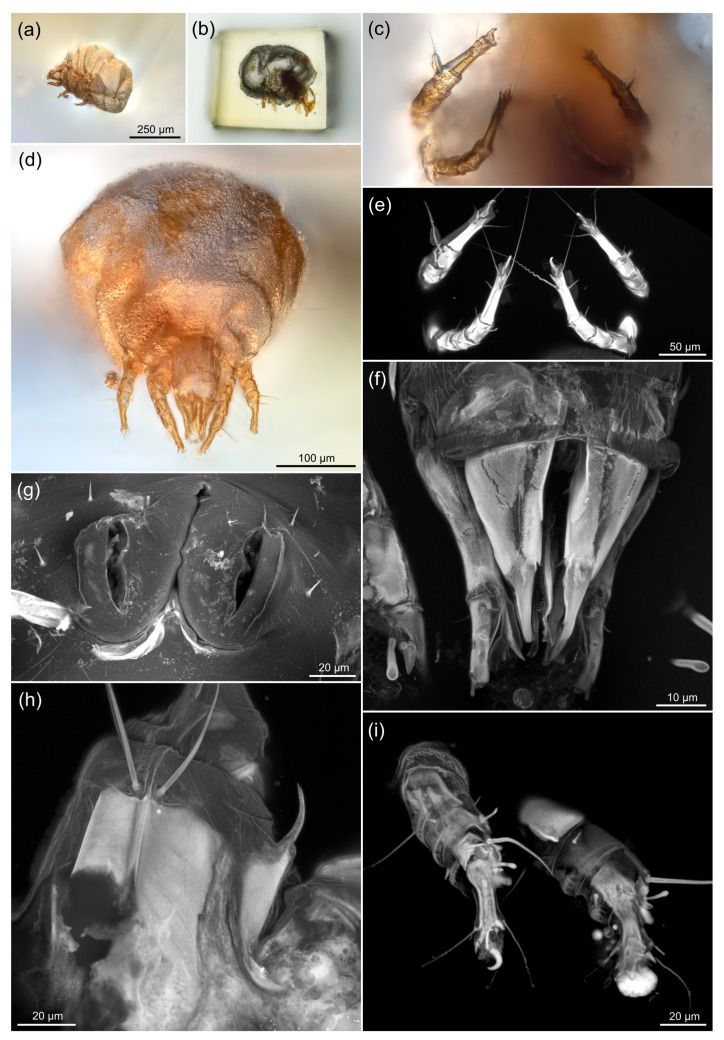
Comparison of traditional and confocal microscopy of the fossil mite. (**a**) A mite in amber piece prior to final polishing; the mite specimen is situated in a suboptimal position relative to the observation plane; (**b**) The same amber piece after the final trimming and polishing; the mite is aligned with the observation planes. Minimal distance is left between the amber surface and the mite to examine the mite with high-resolution optics; (**c**) Ventral view of the legs, transmitted light. The opaque body of the mite significantly obscures details of the legs; (**d**) Frontal view of the mite, transmitted and reflected light; (**e**) Same view as in (**c**), CLSM image. The corkscrew shape of setae probably is an artifact of fossilization, but not of the image processing, as spiral setae are clearly visible in the individual image planes (see the image data at Figshare, link provided below); (**f**–**i**) sCLSM images, showing the details of gnathosoma, palps and legs. Objective magnification: 10× dry (**a**,**b**), 20× dry (**c**,**d**), 40× oil immersion (**e**), 63× oil immersion (**f**–**i**); amber was immersed in saturated fructose solution (**d**–**i**). (**a**–**d**,**i**) Focus-stacking with manual retouching. (**e**–**h**) Maximum intensity projections.

**Figure 4 life-13-00865-f004:**
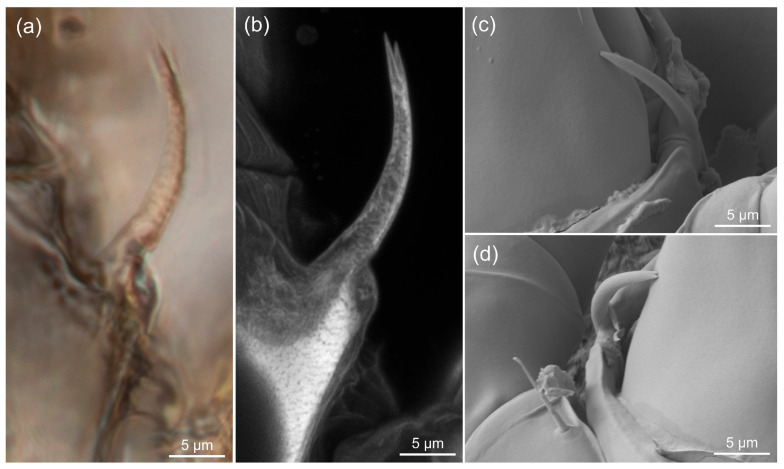
Grandjean’s organ in fossil and modern mites imaged using different methods. (**a**) Transmitted light with differential interference contrast (DIC), 100×/1.30 objective. Semi-automated focus-stacking; (**b**) CLSM, 63×/1.40 objective, 488 nm excitation laser, Airyscan super-resolution detector. Maximum intensity projection; (**c**,**d**) Modern specimens of *Histiogaster arborsignis,* LTSEM. Note the bifurcated tip clearly resolved by CLSM (**b**) and LTSEM (**c**,**d**). All images to same scale.

**Figure 5 life-13-00865-f005:**
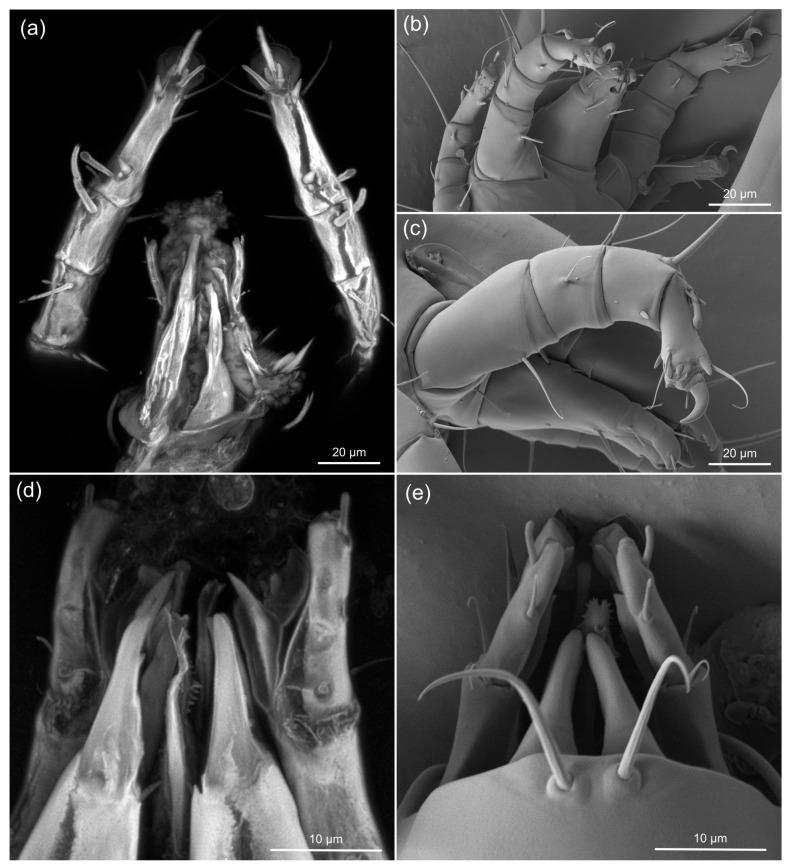
The resolution of sCLSM and LTSEM in Eocene fossil *Histiogaster* mites and modern *Histiogaster* mites, respectively. (**a**,**d**) Gnathosoma and front legs of a fossil. CLSM, 63×/1.40 objective, 488 nm excitation laser, Airyscan super-resolution detector. Maximum intensity projections; (**b**,**c**,**e**) Gnathosoma and front legs of modern specimens of *Histiogaster arborsignis*, LTSEM. (**a**,**b**) and (**c**) to same scale; (**d**) and (**e**) to same scale.

## Data Availability

Image data presented in this study are openly available in FigShare at https://doi.org/10.6084/m9.figshare.22101911 (accessed on 17 February 2023).

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
