# Peer review of "Beyond the Limits of Light: An Application of Super-Resolution Confocal Microscopy (sCLSM) to Investigate Eocene Amber Microfossils"

_life, 2023, doi:10.3390/life13040865_

Round 1

Reviewer 1 Report

I just have a comment and a question:

- for figure 1 (c) it is mentioned microinclusions, possibly fossilized tree sap. These microinclusions do not have the usual characteristics of frequent emulsions in many ambers as described in several articles: for example Lozano et al. (2020) cited by the authors but also Girard et al. (2011), Quinney et al. (2015), Thiel et al. (2016). However, it is difficult to determine if these are bio-inclusions...

- for figure 1 (d), the figure suggests that it is fungal or actinomycetes mycelium. Did the authors test their potential autofluorescence?

Author Response

  • for figure 1 (c) it is mentioned microinclusions, possibly fossilized tree sap. These microinclusions do not have the usual characteristics of frequent emulsions in many ambers as described in several articles: for example Lozano et al. (2020) cited by the authors but also Girard et al. (2011), Quinney et al. (2015), Thiel et al. (2016). However, it is difficult to determine if these are bio-inclusions...

Thank you for this comment, we already argued on this among co-authors. Indeed, the numerous pseudo-inclusions that we found in these two pieces of Eocene amber differed from emulsions of a tree sap described from (mosty) Cretaceous ambers. However, in our specimens all inclusions are at least partially empty, air-filled, unlike the most of Cretaceous (especially Kachin) ambers, and many of them are surrounded by periferal micro-fissures, like in Thiel et al 2016 Fig.1K. Not being sure about the nature of these pseudoinclusions, we just "presumed it to be a fossilized emulsion of a phloem sap". Given the quantity, a range of size (not a stereotyped form) and a more or less uniform distribution in amber, we could not find other plausible explanations, especially in view of the numerous Histiogaster mites found in the same parts of the amber piece. 

Thank you for reminding us about the work of Quinney et al 2015. 

We corrected a phrase in Discussion: "which were tentatively attributed to fossilized phloem sap"

  • for figure 1 (d), the figure suggests that it is fungal or actinomycetes mycelium. Did the authors test their potential autofluorescence?

Yes, we did. Some of the threads indeed demonstrated autofluorescence. Currently we study them and, not being specialists neither in fungi nor in actinomycetes, search for collaboration. To be published, most probably, together with the description of the mites.

Reviewer 2 Report

I enjoyed reading the nicely prepared MS by a group of true scientists from Ukraine, Russia and the US. I agree with almost all points made in this comparative study that showed the efficacy of the application of super-resolution laser confocal microscopy to the study of non-destructive imaging of microscopic fossil animals. As mentioned, a combination of imaging methods such as the super resolution CLSM and μCT used in amber bioinclusion studies exceeds the resolution of commonly used conventional light microscopy. And of course, this proposed imaging method should be widely used by other colleagues who work amber inclusions.     

Minor points:

Line 7: Koltzov Institute of developmental biology, RAS, could this RAS be expanded? We need the full name. Also, capitalize D and B of developmental biology.

Line 43:   Earth evolution, better to say life evolution;

I have also a request:

Is it possible to make a comparison table showing the parameters, weakness and strength, and efficacy of traditional CLSM and sCLSM.

For example, you mentioned 2 steps that are required to achieve the full advantages of sCLSM in amber fossils, but I think you may also need to point out the time cost of each shot under sCLSM. I personally have been asked about the ‘trick’ of taking a nice CLSM image, and I think the timing is a key thing.

Author Response

Thank you for appreciating our work. We hope for wider use of high-resolution imaging in the studies of well-preserved amber fossils – they do deserve it.

Minor points:

Line 7: Koltzov Institute of developmental biology, RAS, could this RAS be expanded? We need the full name. Also, capitalize D and B of developmental biology.

Line 43:   Earth evolution, better to say life evolution;

Thank you, corrected.

I have also a request:

Is it possible to make a comparison table showing the parameters, weakness and strength, and efficacy of traditional CLSM and sCLSM.

 For example, you mentioned 2 steps that are required to achieve the full advantages of sCLSM in amber fossils, but I think you may also need to point out the time cost of each shot under sCLSM. I personally have been asked about the ‘trick’ of taking a nice CLSM image, and I think the timing is a key thing.

There is not much to compare for a separate table. The time cost of imaging with sCLSM was not significantly higher than that of regular CLSM, provided that the pixel dimensions are similar (but magnification and resolution were different, of course).

You are absolutely right, the timing is a key to the image quality. We used about 2μs pixel dwell time and averaged (or summed) 2 frames, which gave ca. 10s for a frame and about 30 min for a scan. It is very convenient that the fossil autofluorescence is rather resistant to bleaching, so both the long pixel time and the high laser intensity may be used. In Airyscan mode, there is also a stage of computation after the scan that takes time, but on our system it was negligible compared to the time of the scanning itself.

We added more details of our method both to the Methods

The parameter that mostly affected the quality of images was the dwell time (or pixel time, the longer the better). We found that auto-fluorescence in amber fossils is resistant to bleaching, so both the long dwell time and the high laser intensity may be used. We normally used dwell time around 2 μs, which gave ca. 10 s frame time for 1000×1000 px, 2 frames averaging, and ca. 30 min scan time for a 150–180 slices stack (examples in Fig.5ad). For sCLSM scans, the z step was usually set to 0.2 μm.

and to the Results:

The time cost of imaging with sCLSM was not significantly higher than that of regular CLSM, provided that the pixel dimensions are similar. For example, scanning of the anterior part of a mite in regular CLSM mode (supplementary video S1) and the mouthparts of the same specimen at higher magnification in Airyscan mode (supplementary video S2) took ca. 30 min in both cases.

Reviewer 3 Report

Very interesting high-quality paper describing a new application of sCLSM method for studying ambr inclusions. When authors compare different methods, they are recommended to show images of the same stractures at exactly the same scale.

Author Response

Thank you for appreciating our work. In all figures with comparison of different imaging methods (Fig.4, Fig.5) we used exactly the same scale for different methods. The scale in the SEM images may seem different because the modern mites were smaller than the fossil ones.

Reviewer 4 Report

This manuscript is an outstanding contribution that will bring great value to its audience. It describes a method that, if widely adopted, can significantly improve the quality of palaeontological data to become more equal to the quality of data from extant specimens. For a few suggestions to improve the manuscript please see the comments in the PDF file.

Author Response

Thank you very much for such a detailed review of our study. Below we comment on several selected points (copied here from the PDF). All other corrections accepted without commenting. Also, please find the same comments in the PDF, it may be easier to read, and it contains an image.

1. "light microscopy has been used since the 18th century [9], however, its success largely depends on the opacity of amber specimens"

What about dark field or phase contrast microscopy?

They too depend on opacity of amber, as every other kind of light microscopy. Phase contrast may be effective in thin layers of fine-polished amber (we sometimes use it), but its resolution is also limited by diffraction. DIC may give nice images (again, in thin pieces of amber), but the resolution limit is more or less the same.

2. Helicon Focus

Was it used here to merge the in-focus areas?

Yes, in some cases when maximum intensity projection did not give satisfactory results, Helicon Focus was used for confocal images stacks.

3. "Specimens in 70% ethanol and/or live material were used for Low Temperature

SEM (LTSEM) studies"

freshly killed?

The Cryo-SEM (or LTSEM) is a very helpful system as you capture the material in time, no chemicals, no dry problems, and directly you review the specimens in their natural environment.

4. Can you provide more information on the objective? Especially at these magnifications the optics play a mayor role for image quality.

We added all objectives used in this study to the Methods and mentioned them in the figure legends.

5. Can you somehow exclude that this wave-like pattern is an artefact of image processing? If so, please mention it more explicitly in the caption.

It is very easy to exclude, since spiral setae are clearly visible in the individual image planes, both TL and confocal. Here's an example (a crop from a single source slice):

Sorry that we did not timely activate ('published') the Figshare link, we expected requests for more data to be put there.

Added explicit statement to the caption of the figure:

The corkscrew shape of setae probably is an artifact of fossilization, but not of the image processing, as spiral setae are clearly visible in the individual image planes (see the image data at Figshare, link provided below);

It is still a mystery to us what exactly causes this effect. First noticed in another Eocene fossil mite (Khautov et al (2021) Systematic & Applied Acarology26(1): 33–61), also from Rovno amber, these screw setae unexpectedly (and rarely) appear along with other, normally straight ones.

6. amber deterioration.

Added more to Intro and to Discussion on this topic.

Interestingly, after the lengthy scans with high intensities of laser emission, we observed the effect of bleaching in the thickness of amber. This, most probably, did not mean that we reversed a depolymerisation of amber by our laser emission. Added this observation to Results.

7. I would also mention here the advantage that the threedimensionality of both the various CT methods and the CLSM offer. This opens possibilities for measurements that would not be possible for amber fossils using light microscopy.

We mentioned about the measurement in 3D a few paragraphs below:

In addition, it is quite easy to represent and measure confocal image stacks in three dimensions (see Supplementary videos S1 and S2).

Added to the Discussion (thank you for the wording):

Three-dimensionality provided by μCT and CLSM opens possibilities for measurements that would be hardly possible for amber fossils using light microscopy

Indeed, we measure structures in 3D using confocal stacks, particularly the setae that project across the scanning plane (not in this study, though). As for the light microscopy, such measurements are not totally impossible. If the microscopic screw (manual or electronic-controlled) is precise enough, similar 3D measurements in stacks can be done, especially using DIC. With 100x objective, the image of amber fossil is quite confocal, the depth of focus is very short.

8. "However, CLSM may serve as a sensitive method for the early diagnostic of the deterioration process. CLSM allows to perform the optical sectioning of the specimen alternative to its physical cutting to demonstrate the gradient of yellowing."

I doubt that this is technically possible or possible without much experimental work and a calibrated workflow. The optical conditions differ the deeper the focus plane is in the amber, and likely also with individual chemical characteristics of the amber pieces. This would need to be accounted for.

We agree. Also the more or less precious time at a good confocal microscope may be another prohibitive obstacle on the way to the mass testing of the collection amber. However, even without a calibration shared between different museums and laboratories, a pairwise comparison of amber pieces can be easily performed: freshly acquired vs stored for some years, epoxy-covered vs raw, etc.

Added to the text:

However, without a calibrated workflow only a pairwise comparison of amber pieces stored in different conditions can be performed.

9. Please check again: Is resilin the main source of fluorescence?

Aparently, it has to be excited with a 405 nm laser to be well visible. Which would suggest to me that since you used a longer wavelength, that your fluorescence is based on other substances. This also opens up the question of how long such proteins can survive in amber

Here we mentioned resilin as only one example (though, the best-studied) of autofluorescent components of the arthropod cuticle. Chitin or chitin-based compounds fit better to the fossil autofluorescence, but still not perfectly. Quite probably, the fluorescence in amber fossils is based on other substances, and perhaps even on the ones that are absent in the cuticle of living animals. This requires a separate dedicated study, which we are already performing. Here we are not discussing what is the nature of the fossil autofluorescence: we just say the "the nature of fluorescence in the fossil exoskeletons remains unclear".

To avoid the direct comparison of the fossil autofluorescence with that of resilin in modern arthropods, we modified the sentence in the Discussion:

In modern arthropods, this effect is based on the presence of resilin or chitinous compounds, with their respective blue or green-to-red autofluorescence, as well as of other fluorescent components [Croce Scolarli 2022].

In the Methods:

Endogenous fluorophores known for insect cuticle, include resilin [34], chitin and chitinous compounds [Rabasović et al 2018] and other substances [Croce Scolarli 2022].

10. to represent and measure confocal image stacks in three dimensions

but what you have in the supplementary files are 2D animations

I'm afraid that we differently understand the term 'three dimensions'. In our view, a 3D representation (on a 2D screen) is a series of projections, controlled by a viewer (who rotates the object around one of the three axes) or a video that successively shows such rotation. These projections can include ether the simulated surfaces (like it is often done for CT scans) or the source intensity data, like it is usually done for confocal scans based on fluorescence. In both cases the and the software operates voxels (x,y,z) instead of pixels, as in 2D animation. Confocal data can be presented in the same way as the CT scans (we use Drishti software for it in another project), but in this case we chose the minimally altered representation of the source stacks with the 3D projection calculated in FIJI (Image...Stacks...3D Project). 2D animation in this case would be the scrolling through the stack along its z axis.

11. In this paragraph you are mixing up synchrotron micro CT and x-ray micro CT. With synchrotron scanning it is possible to use phase contrast, which allows to get good results even for amber fossils that are not hollow. See e.g.

https://doi.org/10.1017/S1431927608080264

or https://doi.org/10.26879/1129

Thank you, we separated the citations of studies based on micro-CT and SRμCT. We agree that synchrotron with phase contrast currently provides better results than a compact micro-CT device. However, if one looks at any of the recently published images of amber fossils prepared using synchrotron (Baranov et al 2021, Perreau et al 2021) and compares them with either the light microscopic images or the SEMs of a similar Extant organism, the fossils look... shaved. This effect may be partially due to the method of calculation of a 3D surface, which always includes a threshold and cuts out setae and other thin projections together with noise. Not aiming at the synchrotron, we are currently testing the abilities of nano-CT scanners in application to fossil mites. Not impressed yet.
